EMBO
*reports*

# Adherens junction remodelling during mitotic rounding of pseudostratified epithelial cells

Mario Aguilar-Aragon[1], Teresa T Bonello[2], Graham P Bell[1], Georgina C Fletcher[1] & Barry J Thompson[1,2,*] (iD)

## Abstract

Epithelial cells undergo cortical rounding at the onset of mitosis to enable spindle orientation in the plane of the epithelium. In cuboidal epithelia in culture, the adherens junction protein E-cadherin recruits Pins/LGN/GPSM2 and Mud/NuMA to orient the mitotic spindle. In the pseudostratified columnar epithelial cells of *Drosophila*, septate junctions recruit Mud/NuMA to orient the spindle, while Pins/LGN/GPSM2 is surprisingly dispensable. We show that these pseudostratified epithelial cells downregulate E-cadherin as they round up for mitosis. Preventing cortical rounding by inhibiting Rho-kinase-mediated actomyosin contractility blocks downregulation of E-cadherin during mitosis. Mitotic activation of Rho-kinase depends on the RhoGEF ECT2/Pebble and its binding partners RacGAP1/MgcRacGAP/CYK4/Tum and MKLP1/KIF23/ZEN4/ Pav. Cell cycle control of these Rho activators is mediated by the Aurora A and B kinases, which act redundantly during mitotic rounding. Thus, in *Drosophila* pseudostratified epithelia, disruption of adherens junctions during mitosis necessitates planar spindle orientation by septate junctions to maintain epithelial integrity.

**Keywords** cortex; E-cadherin; mitosis; myosin
**Subject Categories** Cell Adhesion, Polarity & Cytoskeleton; Development

## Introduction

Epithelial tissues are comprised of highly polarised cells with distinct apical and basolateral plasma membrane domains, separated by a prominent belt of adherens junctions [1–4]. Adherens junctions are composed of E-cadherin (E-cad), beta-catenin/Armadillo (Arm) and alpha-catenin, which recruits an actomyosin contractile ring to maintain precise control of epithelial cell shape [5,6]. In proliferating epithelia, cells must orient their mitotic spindle in the plane of the epithelium to propagate cell polarity through cell division and thus maintain orderly packing of epithelial monolayers [7]. Rounding up of the cell cortex during mitosis is fundamentally important to enable correct formation and orientation of the mitotic spindle by molecular and mechanical cues [8–13].

Key molecules linking the mitotic spindle to the cell cortex in epithelia are the proteins Pins/LGN/GPSM2 and its binding partner Mud/NuMA [14–19]. In *Drosophila*, the basolateral determinants Dlg and Scrib were discovered to function as lateral cues for planar spindle orientation [20,21], while Lgl is removed from the basolateral membrane and becomes cytoplasmic during mitosis [22,23]. In the *Drosophila* ovarian follicle cell epithelium, lateral Dlg binds to Pins/LGN/GPSM2 and Mud/NuMA to orient the mitotic spindle [20]. Binding of Dlg to Pins occurs via the same domain as Dlg-Lgl binding, which may explain why Lgl must be removed from the membrane to orient the spindle in follicle cells [22,23].

In the *Drosophila* wing imaginal disc epithelium, Dlg and Scrib concentrate at septate junctions and directly recruit Mud/NuMA, while Pins/LGN/GPSM2 is dispensable for spindle orientation [21,22,24]. Mud/NuMA also tends to concentrate at tricellular junctions in a Gliotactin-dependent manner, but Gliotactin is not required for planar spindle orientation [25]. Notably, the wing imaginal disc is a pseudostratified columnar epithelium, such that mitotic rounding occurs at the apical surface and coincides with apical movement of the nucleus, a broadly conserved process known as interkinetic nuclear migration [26–29]. In the absence of mitotic rounding in pseudostratified epithelia, the spindle fails to be correctly oriented by planar cues and can instead orient aberrantly in the apical–basal axis, leading to extrusion of daughter cells from the epithelium and subsequent apoptosis [21,29,30].

Mitotic rounding is known to require uniform activation of Myosin-II-mediated cortical contractility by the Rho GTPase and its effector Rho-kinase (Rok/ROCK) [21,26,31–33]. Mitotic activation of Rho occurs in response to activation of the cell cycle-regulated GTP exchange factor (GEF) Pebble (Pbl/ECT2) [34–39]. In addition, mitotic rounding involves activation of the ERM protein Moesin to promote attachment of the actin cytoskeleton to the plasma membrane and ensure proper spindle orientation [40–42]. Finally, the adherens junction protein beta-catenin/Arm was reported to be degraded during mitosis in *Drosophila* [43], while, in contrast, adherens junctions are retained in some cultured mammalian epithelial cells during mitosis, where they can recruit Pins/LGN/GPSM2 to

1 Epithelial Biology Laboratory, Francis Crick Institute, London, UK
2 EMBL Australia, The John Curtin School of Medical Research, The Australian National University, Acton, ACT, Australia
*Corresponding author. Tel: +61 4368 38584; E-mail: barry.thompson@anu.edu.au

orient the spindle [44–46]. Thus, whether and how adherens junctions are controlled during mitosis in different epithelial tissue types is still unclear.

Here, we show that E-cadherin is indeed downregulated in the pseudostratified wing epithelium of *Drosophila*. We propose that loss of adherens junctions is mechanically induced in pseudostratified epithelial cells. The tall and narrow shape of these cells in interphase necessitates a particularly extensive shape change upon mitotic cell rounding driven by cell cycle-regulated actomyosin contractility. We delineate the molecular mechanisms linking the core cell cycle machinery with mitotic rounding, which is mediated by Aurora A/B kinase activation of the RhoGEF ECT2/Pbl (*Drosophila* Pebble; Pbl) via its binding partners RacGAP1/MgcRacGAP/CYK4/Tum (*Drosophila* Tumbleweed; Tum) and MKLP1/KIF23/ZEN4/Pav (*Drosophila* Pavarotti; Pav). Finally, loss of adherens junctions may explain the necessity for spindle orientation by septate junctions in these pseudostratified cells, while other cell types that retain adherens junctions through mitosis can employ them directly in spindle orientation.

## Results

### Epithelial cells round up and downregulate adherens junctions at mitosis

We first examined the localisation of fluorescently tagged forms of the adherens junction proteins Armadillo (Arm) and E-cadherin (E-cad) by live imaging. We find that both Arm-GFP and E-cad-GFP are downregulated as cells round up during mitosis (Fig 1A–C). Following cytokinesis, Arm-GFP and E-cad-GFP re-form a prominent belt adherens junctions as the cells return to their normal shape (Fig 1A and B). When cells are arrested in mitosis with colchicine, the weakened belt of adherens junctions never returns to normal levels (Fig 1C). Quantifications show that the levels of both Arm-GFP and E-cad-GFP are reduced by approximately 50% at the junctions between mitotic cells and their interphase neighbours (Fig 1D). Much of this residual 50% appears to come from the neighbouring cells, rather than the mitotic cell itself, since no detectable Arm-GFP signal can be detected at the interface of two adjacent cells that happen to enter mitosis at the same time (Fig 1E). Electron microscopy confirms that adherens junctions are visible in interphase and prophase cells, but only weakly present in prometaphase and telophase cells (Fig 1F–I). These results show that adherens junctions are transiently downregulated during mitosis, presumably in order to accommodate the extensive rounding up of these pseudostratified epithelial cells at this point in the cell cycle.

### Rho-kinase activates actomyosin contractility to induce downregulation of adherens junctions at mitosis

In interphase cells, moderate actomyosin contractility stabilises adherens junctions while increased activation of actomyosin contractility has been reported to be sufficient to cause endocytic downregulation of E-cadherin [47,48]. To test whether the increase in actomyosin contractility during mitotic cell rounding causes the downregulation of adherens junctions during mitosis, we sought to block actomyosin contractility during mitosis. Phosphorylation of

Myosin-II regulatory light chain (Myo-II RLC or Squash, Sqh) is known to activate actomyosin contractility [49,50]. During mitosis, a pathway from cyclin-dependent kinase 1 (CDK1) to activation of Rho-kinase (Rok), which phosphorylates Myo-II RLC, has been implicated in mitotic rounding [26,33,34]. Silencing of Rho-kinase by expression of *UAS.Rok-RNAi* in the posterior compartment of the wing imaginal disc impaired both the mitotic phosphorylation of Myo-II RLC (p-MLC or p-MyoII) and loss of E-cadherin (Fig 2A–C). Conversely, overexpression of constitutively active Rho-kinase (*UAS.Rok-CA*) in the posterior compartment of the wing imaginal disc uniformly elevated phosphorylation of Myo-II RLC and downregulated E-cadherin levels across the compartment (Fig 2D). We then used the Rho-kinase inhibitor Y-27632 to acutely block mitotic phosphorylation of Myo-II RLC and accumulation of Myo-II RLC-GFP at the cortex in mitotic wing epithelial cells, which also prevented the mitotic downregulation of E-cad-GFP in the wing epithelium (Fig 2E and F). Notably, extended treatment of the epithelium with Y-27632 for more than an hour caused profound defects in interphase cells, with apical surface area enlarging and adherens junctions being generally downregulated (Fig 2F). These results show that adherens junction localisation normally requires an optimal level of actomyosin contractility during interphase and that the mitotic activation of contractility during rounding leads to loss of adherens junctions.

We further tested whether ectopic activation of the Rho GTPase itself is sufficient to downregulate adherens junctions, even in interphase cells. To activate Rho, we overexpressed a constitutively active form of the protein, RhoV14, in GFP-marked clones of cells with the *GAL4 UAS* system. We find that GFP-positive cells round up and exhibit increased cortical F-actin while at the same time having decreased E-cadherin levels at the plasma membrane (Fig EV1A and B). As expected, epithelial morphology is completely abnormal in these mutant clones (Fig EV1B). These results confirm that ectopic activation of Rho-mediated actomyosin contractility is both necessary and sufficient for cell rounding and consequent downregulation of adherens junctions at mitosis.

### The RhoGEF Pebble/ECT2 activates Rho to downregulate adherens junctions

Previous work has identified a key role for the RhoGEF ECT2/Pebble (Pbl) in activating the RhoGTPase during mitotic rounding and cytokinesis in the *Drosophila* embryo and notum as well as in mammalian cells [34–39]. These findings predict that Pbl might also be responsible for Rho activation during mitosis in the pseudostratified wing epithelium of *Drosophila*. In support of this view, release of Pbl from the nucleus into the cytoplasm at mitosis correlates with re-localisation of GFP-tagged Rok to the plasma membrane (Fig 3A). Furthermore, expression of UAS.Pbl-RNAi prevents mitotic accumulation of p-MyoII in the wing epithelium (Fig 3B). Given the well-established importance of cell rounding for planar spindle orientation, we expected that loss of Pbl might lead to mis-orientation of the spindle and thus extrusion of cells from the epithelium and apoptosis, or alternatively to a complete failure cytokinesis. We find that expression of *ptc.Gal4 UAS.Pbl-RNAi* in a stripe of cells in the wing disc leads to apoptosis of cells as well as enlargement of those cells that remain in the epithelium, a classic consequence of cytokinesis failure (Figs 3C and EV2). We note that the extrusion and

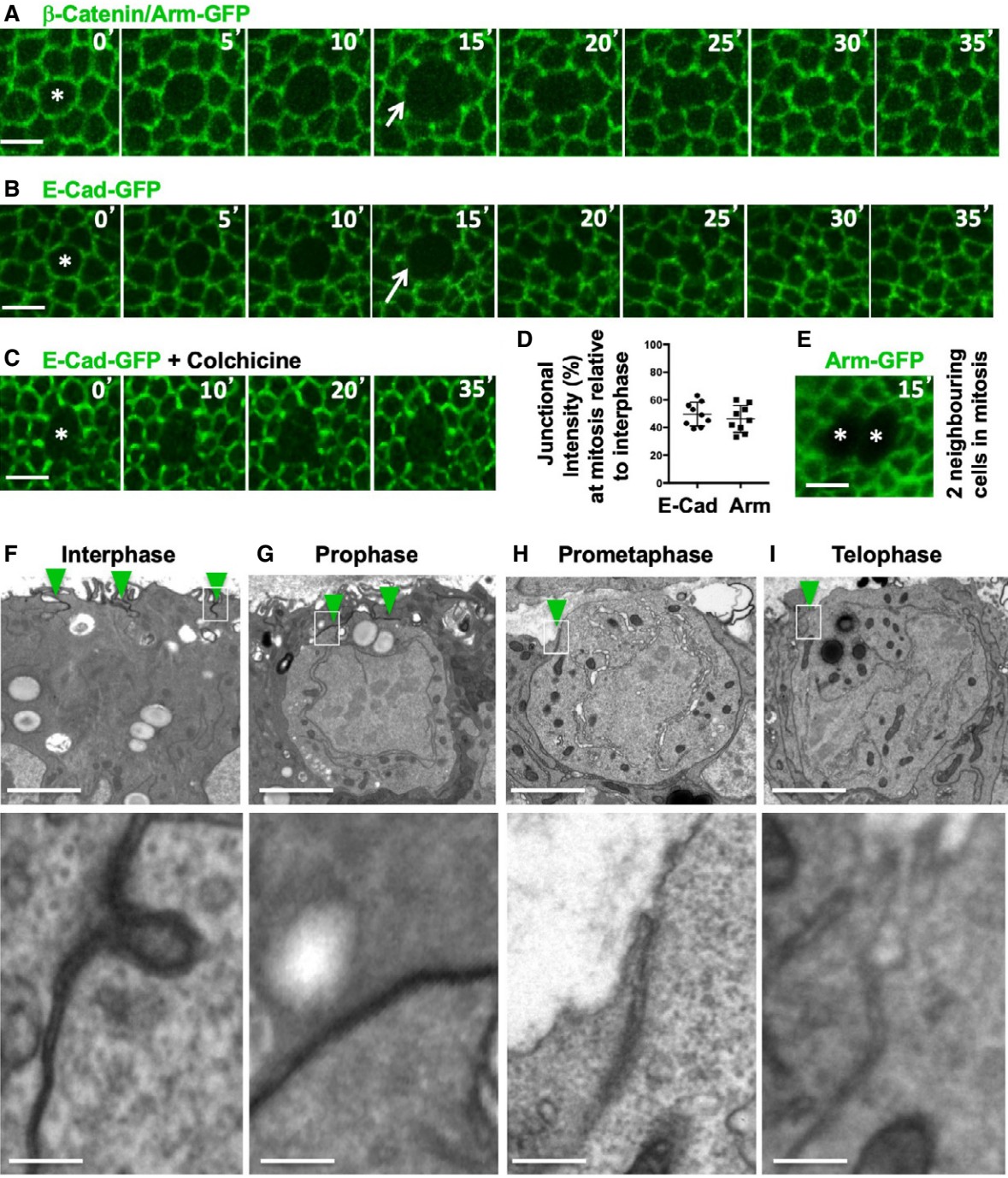

**Figure 1. Epithelial cells round up and downregulate adherens junctions at mitosis.**

A   Dynamic regulation of fluorescently tagged beta-catenin/Armadillo (Arm-GFP) during mitosis in the growing fly wing epithelium. Notice downregulation of Arm-GFP as cells round up for mitosis (arrow). Scale bar ~2 μm. *n* > 10 independent biological replicates.

B   Dynamic regulation of fluorescently tagged E-cadherin (E-cad-GFP) during mitosis in the growing fly wing epithelium. Notice downregulation of E-cad-GFP as cells round up for mitosis (arrow). Scale bar ~2 μm. *n* > 10 independent biological replicates.

C   E-cad-GFP remains downregulated in cells arrested in mitosis with colchicine. Scale bar ~2 μm. *n* > 10 independent biological replicates.

D   Quantification of the degree of downregulation of Arm-GFP and E-cad-GFP at the junction between mitotic cells and their interphase neighbours (*n* > 10 independent samples per cell cycle stage). Mean ± 1 SD shown.

E   Two adjacent mitotic cells show no detectable Arm-GFP at the junction between them. Scale bar ~2 μm.

F–I   Electron microscopy cross-sections of wing epithelial cells at interphase, prophase, prometaphase and telophase. Notice adherens junctions (dark staining, highlighted in insets) are downregulated as cells round up for mitosis. Scale bars ~1 μm (low mag.) and ~10 nm (high mag.). *n* > 10 independent biological replicates. Asterisks indicate a single mitotic cell. Green arrowheads indicate adherens junctions.

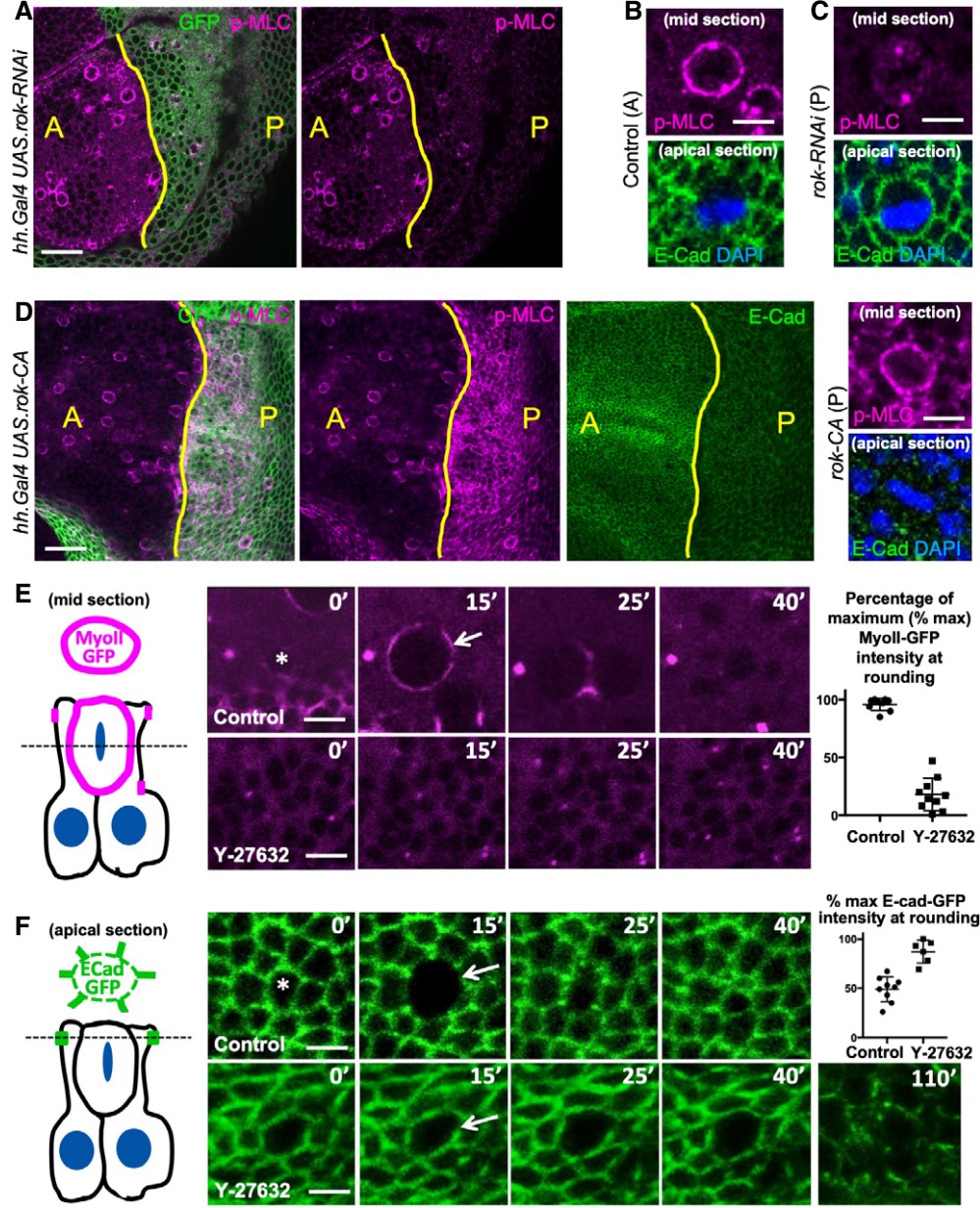

**Figure 2. Rho-kinase activates actomyosin contractility to induce downregulation of adherens junctions at mitosis.**

A   Phosphorylated Myosin-II regulatory light chain (p-MLC; purple) detected by antibody staining reveals high levels around the cortex of rounded mitotic cells in wild-type wing epithelial cells in the anterior (A) compartment, but not in Rho-kinase (Rok) RNAi expressing cells of the posterior (P) compartment (GFP-positive; green). Scale bar ~10 μm. *n* > 10 independent biological replicates.

B   High-magnification view of single mitotic cell stained for p-MLC (purple) and E-cadherin (green), with metaphase plate chromosomes stained by DAPI (blue). Scale bar ~1 μm. *n* > 10 independent biological replicates.

C   High-magnification view of single *UAS.Rok-RNAi* expressing mitotic cell stained for p-MLC (purple) and E-cadherin (green), with metaphase plate chromosomes stained by DAPI (blue). Note loss of cortical p-MLC staining (purple). Scale bar ~1 μm. *n* > 10 independent biological replicates.

D   Expression of constitutively active Rho-kinase (*UAS.Rok-CA*) in the posterior compartment of the wing epithelium is sufficient to elevate p-MLC and reduce E-cadherin immunostaining. Zoom (right) shows high p-MLC and low E-cad in both a mitotic cell and its neighbours. Scale bar ~10 μm. *n* > 10 independent biological replicates.

E   MyoII RLC-GFP (MyoII-GFP) accumulates at the cortex as cells round up for mitosis and later appears at the cleavage furrow during cytokinesis (basal section, diagrammed left). Treatment with the Rho-kinase inhibitor Y-27632 inhibits mitotic accumulation of MyoII-GFP. Quantification shown right (*n* = 8 independent samples per genotype). Scale bar ~1 μm. Mean ± 1 SD shown.

F   E-cad-GFP is downregulated as cells round up for mitosis (apical section, diagrammed left). Treatment with the Rho-kinase inhibitor Y-27632 inhibits mitotic downregulation of E-cad-GFP. Extended treatment with Y-27632 leads to a general defect in all interphase and mitotic cells in which apical surface areas enlarge and E-cad-GFP is generally lost. Arrow indicates mitotic cell cortex at maximal rounding. Quantification shown right (*n* = 6 independent samples per genotype; mean ± 1 SD shown). Scale bar ~1 μm.

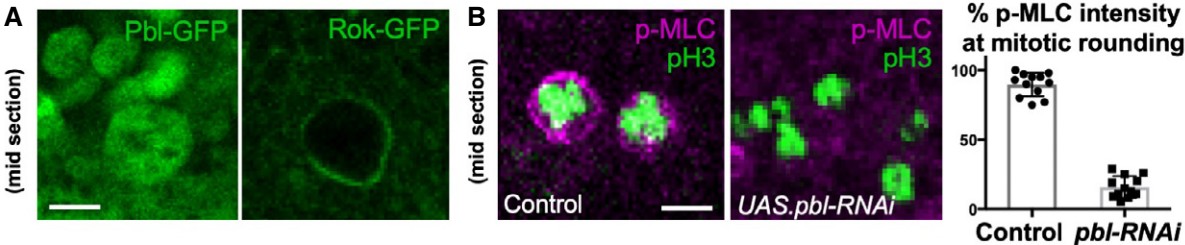

**Figure 3. Loss of RhoGEF ECT2/Pbl prevents mitotic rounding and causes cell extrusion and apoptosis.**

A  Nuclear localised Pbl-GFP becomes cytoplasmic in mitotic cells, co-incident with Rok-GFP re-localisation to the plasma membrane. Scale bar ~1 μm. *n* > 5 independent biological replicates.

B  Phosphorylated Myosin-II regulatory light chain (p-MLC; purple) detected by antibody staining reveals high levels around the cortex of rounded mitotic cells in wild-type wing epithelial cells, but not those expressing *UAS.pbl-RNAi*. Phospho-Histone H3 (pH3) staining (green) marks mitotic cell chromosomes. Quantification shown right (*n* > 10 independent samples per genotype; mean ± 1 SD shown). Scale bar ~1 μm.

C  A stripe of expression of *ptc.Gal4 UAS.pbl-RNAi* in the wing disc leads to both enlarged cells and many basally extruded cells that undergo apoptosis, marked by pyknotic nuclei and loss of aPKC/Dlg staining. Scale bars ~20 μm. *n* > 6 independent biological replicates.

apoptosis phenotype of *UAS.Pbl-RNAi* is highly similar to loss of Moesin [51,52], which is also required for mitotic rounding [40–42], or loss of Mud/NuMA, which is required for planar spindle orientation [21]. Thus, our results confirm that Pbl is indeed essential for Rho activation, mitotic rounding, planar spindle orientation and cytokinesis in the *Drosophila* wing disc.

A key binding partner of ECT2/Pbl is RacGAP1/MgcRacGAP/CYK4/Tum, which has been shown to be essential to activate ECT2/Pbl at the cytokinetic furrow [37,53–55]. Here, we find that GFP-tagged Tum (also called RACGAP50C) localises to the nucleus during interphase—similar to ECT2/Pbl—but is released during mitosis and localises to the cytoplasm and plasma membrane (Fig 4A). The plasma membrane localisation of GFP-Tum during mitosis is relatively less than that found at the cytokinetic furrow or at ring canals formed after cytokinesis, but is nevertheless greater than that occurring during interphase, when almost no GFP-Tum can be detected outside the nucleus or at remnant ring canals (Fig 4A). We further find that expression of *UAS.tum-RNAi* causes a phenotype similar to *UAS.pbl-RNAi* in the wing epithelium (with cell extrusion and apoptosis) and induction of homozygous *tum347* mutant clones prevents accumulation of p-MyoII and cell rounding —notably, only single-cell clones could be obtained, suggesting that mutant cells are eliminated upon division (Fig 4B and C). These findings demonstrate cell cycle regulation of Tum localisation and an essential role for RacGAP1/MgcRacGAP/CYK4/Tum in ECT2/Pbl-mediated Rho activation and actomyosin-driven mitotic rounding in the pseudostratified wing epithelium of *Drosophila*.

### Aurora A and B kinases are required for mitotic rounding and downregulation of adherens junctions

We next sought to examine cell cycle control of Rho activation during mitotic rounding. In *Drosophila*, the rate-limiting molecule for mitotic entry is Cdc25 (String), which dephosphorylates and activates the mitotic CyclinB-Cdk1 complex [56–59]. How Cdk1 then controls Pbl/ECT2 activation for mitotic rounding and subsequent cytokinesis is still not fully understood. One group suggested that Cdk1 directly phosphorylates T412 in Pbl/ECT2 to promote its interaction with Polo-like kinase (Plk1) [38,60]. Others showed that Plk1 acts downstream of Cdk1 to phosphorylate RacGAP1/MgcRacGAP/CYK4/Tumbleweed (Tum), which creates a phospho-epitope recognised by the BRCT repeats of Pbl/ECT2 to induce a conformational change and activation [61,62]. How Cdk1 activates Plk1 is still unclear, but the Aurora kinases are possible intermediaries, as Cdk1 phosphorylates the Aurora binding partner Bora [63] (also called SPAT-1) to promote Aurora A activation and phosphorylation of Plk1 in *C. elegans* and human cells [64,65]. However, *Aurora A* (*AurA*) mutants are homozygous viable throughout larval development, do not prevent mitotic entry and instead only cause mild delays in mitotic entry, spindle assembly and asymmetric cell division [63,66–68], while Aurora B loss of function primarily affects cytokinesis [69–71] and may act in parallel with Plk1 by directly [72] phosphorylating the kinesin MKLP1/KIF23/ZEN4/Pav [73,74], which enables it to undergo oligomeric clustering with RacGAP1/MgcRacGAP/CYK4/Tum at the spindle midzone [75,76] and nearby plasma membrane [77,78] to promote Pbl/ECT2 recruitment to the plasma membrane [79,80] and Rho-driven cleavage furrow formation in cytokinesis [81]. Thus, while much is known

about the molecular mechanisms governing Pbl/ECT2 and Rho-mediated contractility by Aurora B and PLK1 via the central spindle in cytokinesis, our understanding of the upstream cell cycle regulation of Pbl/ECT2 and Rho during mitotic rounding is far more limited.

We find that the Aurora A and B kinases act redundantly to control mitotic rounding and downregulation of adherens junctions in the *Drosophila* wing epithelium, as inhibiting both kinases with the pan-Aurora inhibitor VX-680 is necessary to prevent rounding and disruption of junctions (Fig 5A–C). Loss of Aurora A alone (in *aurA* mutants) causes delayed mitosis, with cells becoming trapped in a rounded state, with uniform cortical p-MyoII, for long periods (Figs 5A and B, and 6A and B). Loss of Aurora B alone (with *aurB-RNAi*) does not affect rounding but prevents cytokinesis (Figs 5C and 6C). Expression of *aurB-RNAi* in *aurA* mutants causes a much more pronounced phenotype, arresting cell division and/or causing cell extrusion and apoptosis (Fig 6A–G). These results suggest that acute chemical inhibition of both Aurora A and B prevents mitotic rounding and junctional downregulation, while genetic inactivation of both Aurora A and B either completely prevents mitotic progression or, when mitosis has already initiated, leads to extrusion and apoptosis—similar to spindle mis-orientation mutants.

To further investigate acute chemical inhibition of Aurora A and B, we treated cells that had already initiated mitotic rounding with VX-680. We find that rounding cells (marked by cortical accumulation of MyoII-GFP) immediately arrest mitotic progression and remain in a partially rounded state rather than undergoing cytokinesis (Fig 7A and B). Such rounded cells are also unable to complete their downregulation of adherens junctions (Fig 7C). These results suggest a model in which there is a continuous requirement for both Aurora A and B activity for mitotic rounding, junctional downregulation and progression to cytokinesis (Fig 7D).

Finally, we compared our results in pseudostratified epithelia with the cuboidal cells of the *Drosophila* ovarian follicle cell epithelium. The cuboidal shape of follicle cells means that these cells undergo a much less extensive shape change during mitotic rounding. Consequently, we hypothesise that the degree of mechanical strain experienced by the cell cortex is relatively lower in follicle cells, and E-cadherin-based adherens junctions are not downregulated during mitotic rounding, despite increased mechanical stress caused by increased Myosin-II contractility (Fig 8A–C). These results indicate that downregulation of E-cadherin occurs only above a critical degree of cell shape change (a strain threshold), which is uniquely important to the extensive rounding occurring in pseudostratified epithelia.

## Discussion

Adherens junctions have long been thought to be continuously essential for maintaining epithelial form and function. Our findings demonstrate transient loss of adherens junctions during division of pseudostratified epithelial cells, an event that involves adherens junction remodelling during the extensive rounding up of cell shape in mitosis. We have furthermore shown that loss of adherens junctions is a direct consequence of the increased Rho activity and actomyosin contractility that drives mitotic rounding, which is both necessary and sufficient to regulate the level of junctional

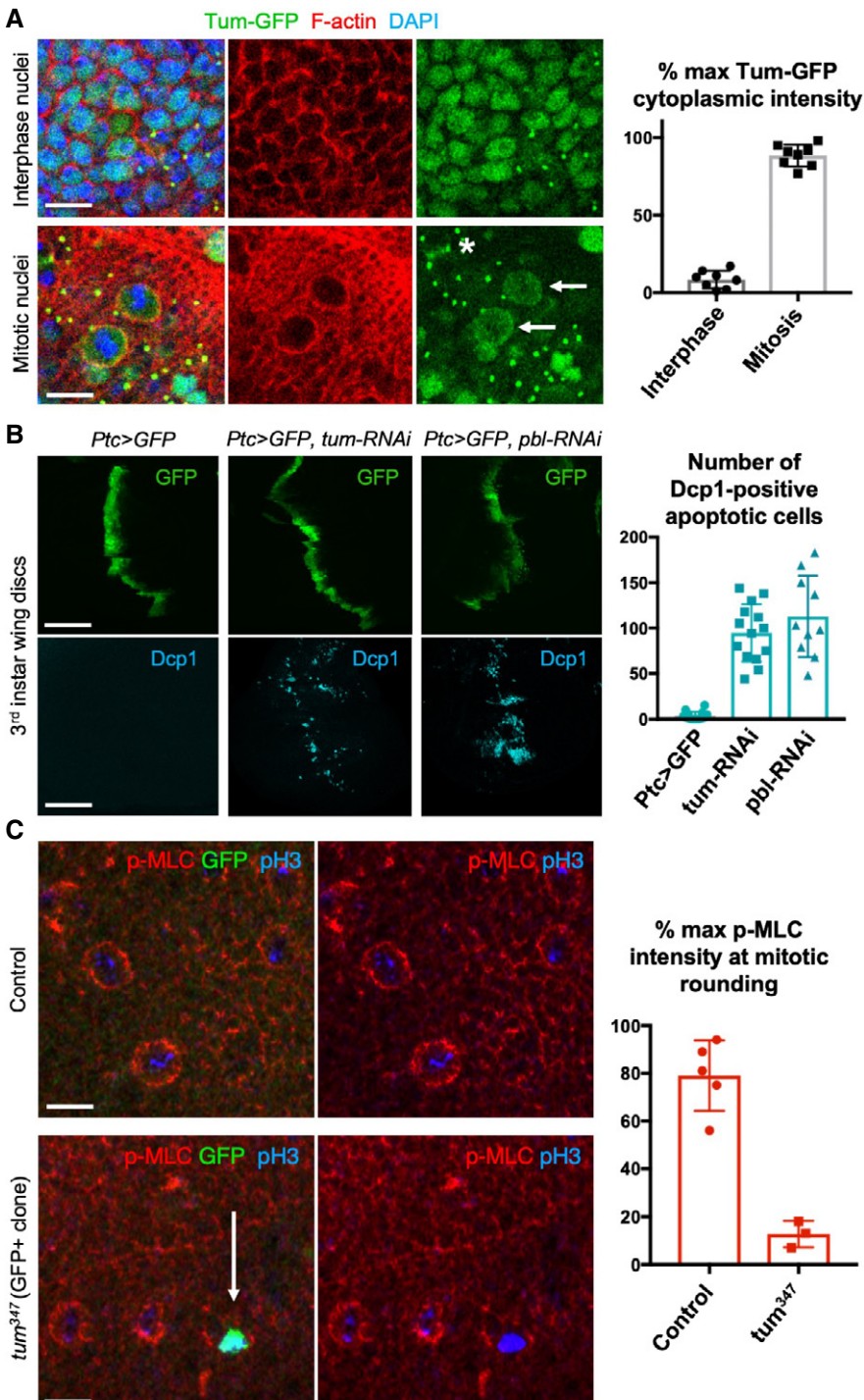

**Figure 4.  Loss of RacGAP1/MgcRacGAP/CYK4/Tum prevents mitotic rounding and causes cell extrusion and apoptosis.**

A  Nuclear localised Tum-GFP becomes cytoplasmic in mitotic cells, including a distinct plasma membrane localisation (arrows). Note also localisation to the cytokinetic ring (asterisk) and remnant mid-bodies/ring canals in interphase cells. Quantification shown on the right (*n* > 7 independent samples per genotype; mean ± 1 SD shown). Scale bars ~2 μm.

B  A stripe of expression of *ptc.Gal4 UAS.tum-RNAi* in the wing disc leads to basally extruded cells that undergo apoptosis, marked by Dcp1 immunostaining, similar to *UAS.pbl-RNAi*. Quantification shown on the right (*n* > 10 independent samples per genotype; mean ± 1 SD shown). Scale bars ~20 μm.

C  Phosphorylated Myosin-II regulatory light chain (p-MLC; red) detected by antibody staining reveals high levels around the cortex of rounded mitotic cells in wild-type wing epithelial cells, but not those cells homozygous mutant for *tum*[347], marked by expression of GFP in single-celled clones (MARCM system). pH3 staining (blue) marks mitotic cell chromosomes. Arrow indicates a GFP-positive single-cell *tum*[347] mutant clone. Quantification shown on the right (*n* > 3 independent samples per genotype; mean ± 1 SD shown). Scale bars ~20 μm.

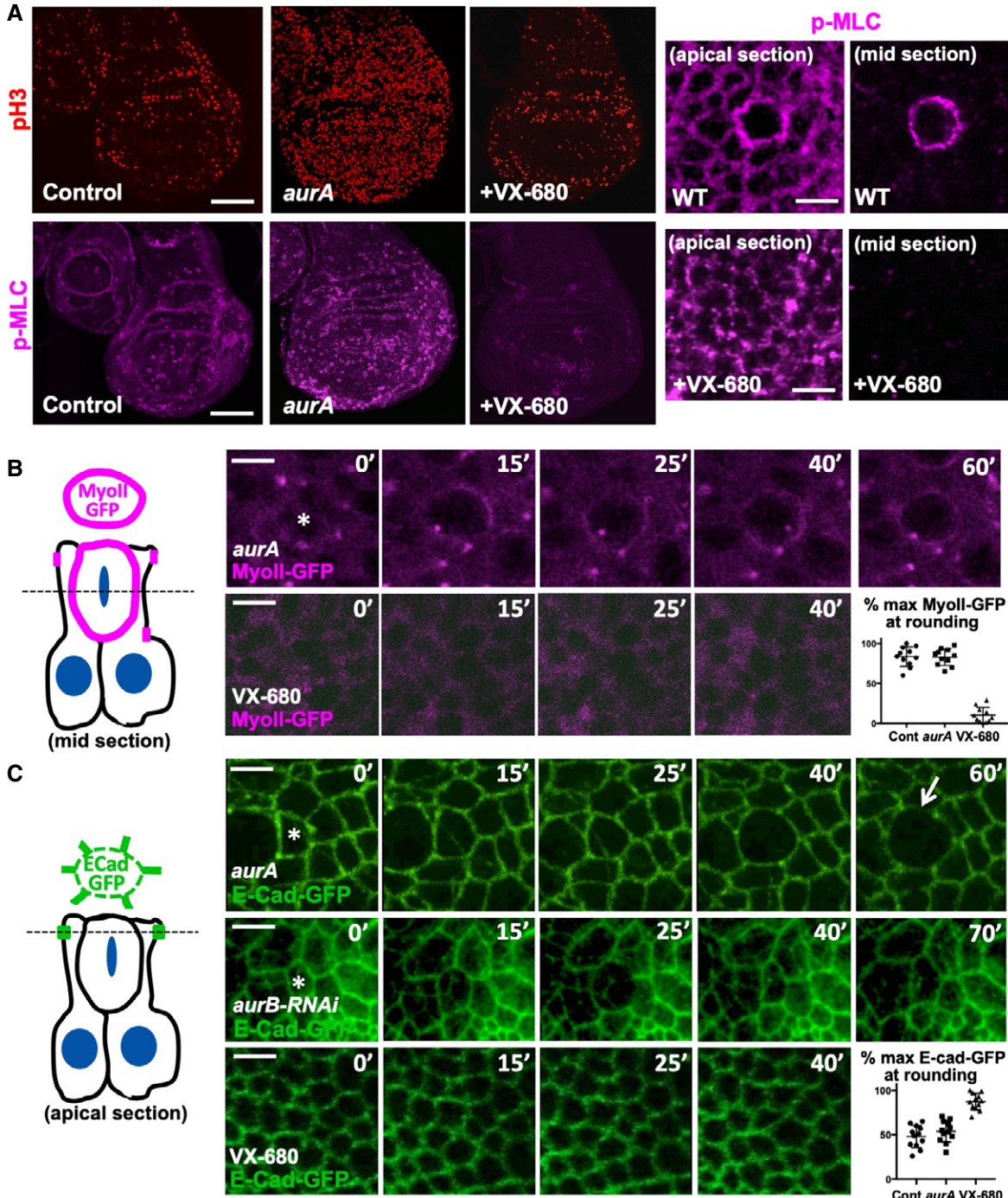

**Figure 5. Mutation of *Aurora A* traps rounded cells in a delayed and extended mitosis, while chemical inhibition of both Aurora A and B prevents mitotic rounding.**

A  Phosphorylated Myosin-II regulatory light chain (p-MLC; purple) detected by antibody staining reveals high levels around the cortex of rounded mitotic cells (pH3, red) in wild-type wing discs, and *aurA* homozygous mutant wing discs, but not upon inhibition of both AurA and AurB by treatment with VX-680. Note increased numbers of mitotic cells in *aurA* homozygous mutant wing discs, due to delayed mitotic progression. Scale bars ~20 μm (low mag.) and ~1 μm (high mag.). *n* > 15 independent biological replicates.

B  Live imaging of MyoII-GFP reveals delayed mitosis in *aurA* homozygous mutant wing discs, while inhibition of both AurA and AurB by treatment with VX-680 leads to complete loss of mitotic rounding and MyoII-GFP membrane localisation. Asterisks indicate a single mitotic cell. Scale bars ~1 μm. Quantification shown on the bottom right (*n* = 5 independent samples per genotype).

C  Live imaging of E-cad-GFP reveals delayed mitosis in *aurA* homozygous mutant wing discs, cytokinesis failure in *UAS.aurB-RNAi* expressing wing discs, and complete failure of mitotic rounding and E-cad-GFP downregulation upon inhibition of both AurA and AurB by treatment with VX-680. Asterisks indicate a single mitotic cell. Scale bars ~1 μm. Quantification shown on the bottom right (*n* = 9 independent samples per genotype).

                                                      

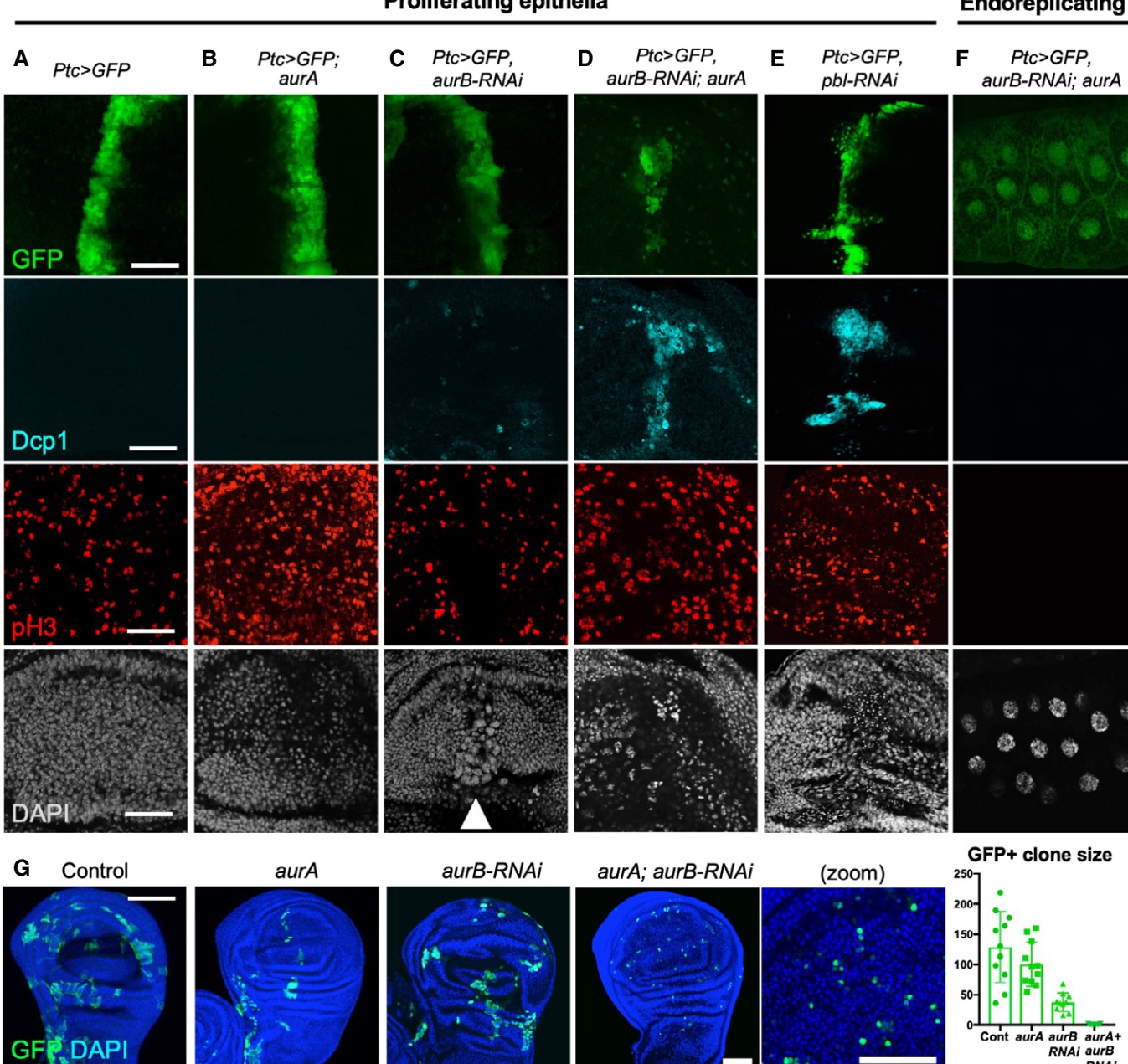

**Figure 6. Genetic redundancy between Aurora A and B kinases *in vivo*.**

A  Striped *ptc.Gal4*-driven expression of *UAS.GFP* in a control wing imaginal disc. Scale bars ~20 μm. *n* > 6 independent biological replicates.

B  Striped *ptc.Gal4*-driven expression of *UAS.GFP* in an *aurA* homozygous mutant wing imaginal disc leads to increased numbers of pH3-positive mitotic cells. *n* > 7 independent biological replicates.

C  Striped *ptc.Gal4*-driven expression of *UAS.GFP* and *UAS.aurB-RNAi* in a control wing imaginal disc leads to enlarged cells with large nuclei (marked by DAPI; white arrowhead). *n* > 6 independent biological replicates.

D  Striped *ptc.Gal4*-driven expression of *UAS.GFP* and *UAS.aurB-RNAi* in an *aurA* homozygous mutant wing imaginal disc leads to extrusion and apoptosis of cells (marked by Dcp1 and pyknotic nuclei). *n* > 8 independent biological replicates.

E  Striped *ptc.Gal4*-driven expression of *UAS.GFP* and *UAS.pbl-RNAi* in the wing imaginal disc leads to extrusion and apoptosis of cells (marked by Dcp1 and pyknotic nuclei). *n* > 8 independent biological replicates.

F  *ptc.Gal4*-driven expression of *UAS.aurB-RNAi* in an *aurA* homozygous mutant endoreplicating salivary gland has no phenotypic consequence. *n* > 6 independent biological replicates.

G  MARCM clones (GFP-positive) expressing *UAS.aurB-RNAi* in an *aurA* homozygous mutant cells result in single-cell clones (i.e. no cell division), while individual loss of AurA or AurB allows some cell proliferation. Quantification shown on the right (*n* > 7 independent samples per genotype; mean ± 1 SD shown). Scale bar ~20 μm.

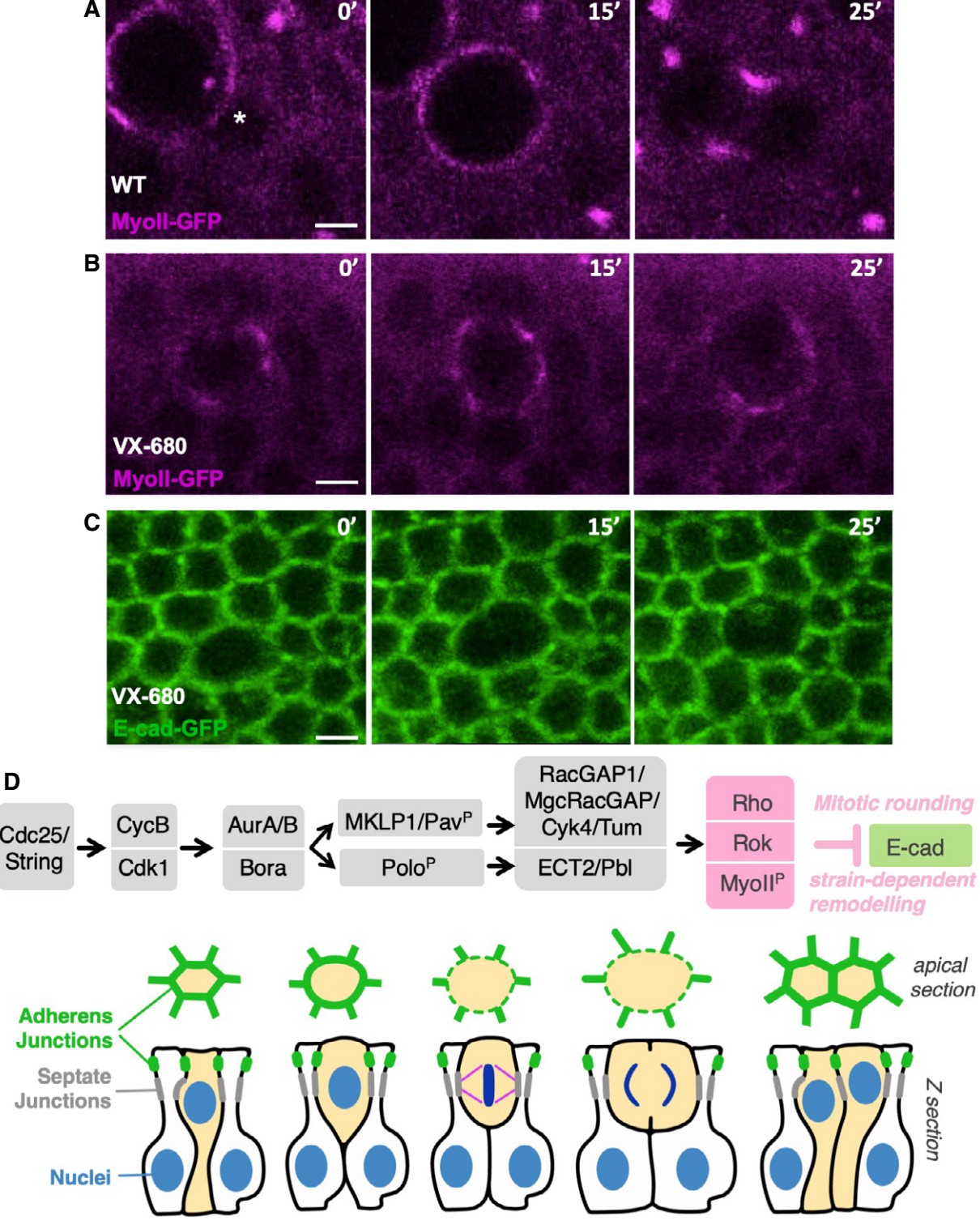

**Figure 7. Aurora A and B are continuously required for mitotic rounding and subsequent cytokinesis.**

A   Live imaging of MyoII-GFP in a control wing disc epithelium. Asterisk indicates the location of a single mitotic cell about to initiate rounding. Scale bar ~1 μm. n > 5 independent biological replicates.

B   Live imaging of MyoII-GFP upon treatment with the dual AurA/B inhibitor VX-680 after initiation of mitotic rounding, which rapidly arrests progression of rounding and prevents cytokinesis. Scale bar ~1 μm. n > 9 independent biological replicates.

C   Live imaging of E-cad-GFP upon treatment with the dual AurA/B inhibitor VX-680 after initiation of mitotic rounding, which rapidly arrests progression of rounding and prevents cytokinesis. Scale bar ~1 μm. n > 6 independent biological replicates.

D   Working model for mitotic rounding and junctional downregulation in pseudostratified *Drosophila* epithelial cells.

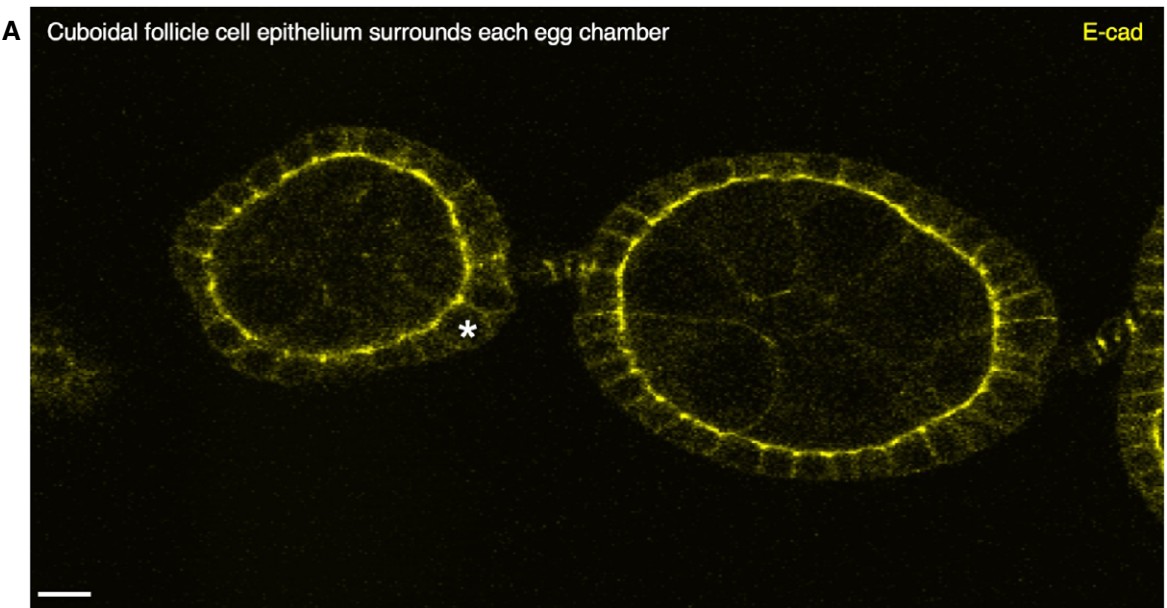

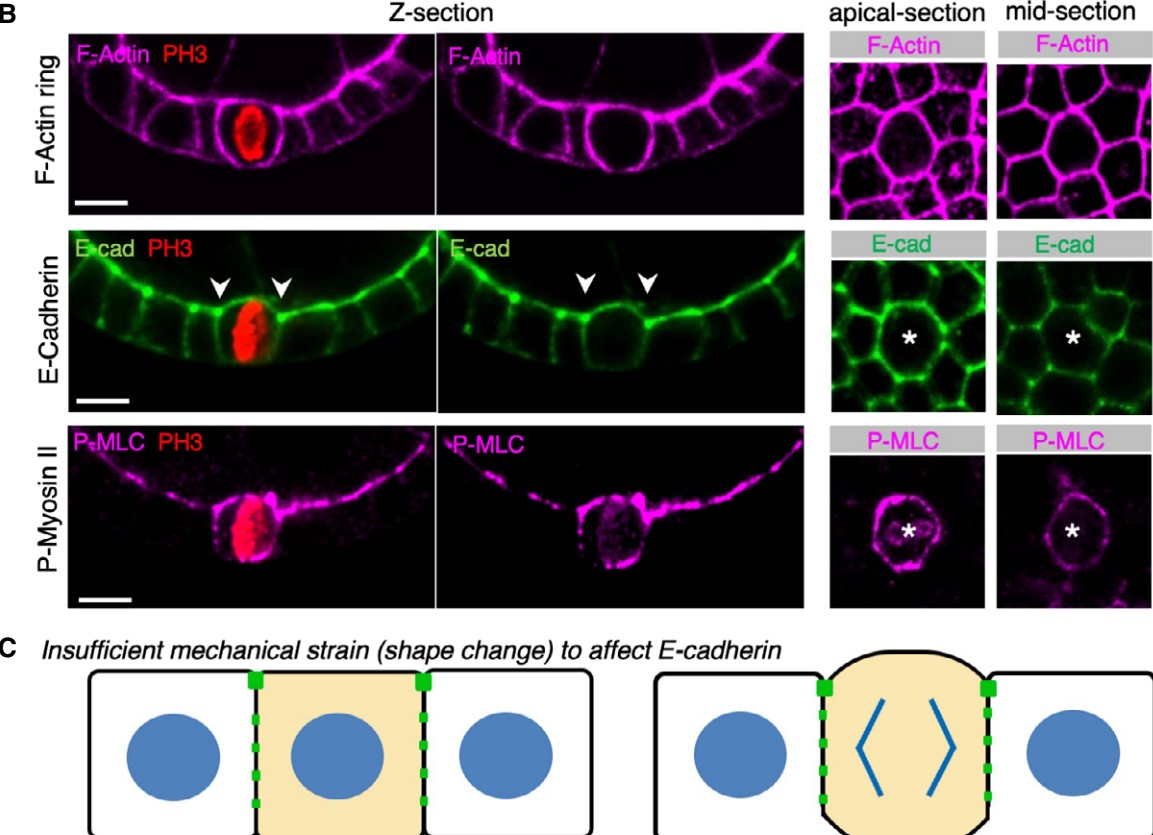

**Figure 8. The cuboidal follicle cell epithelium does not downregulate E-cadherin at mitosis.**

A   Early stage *Drosophila* egg chambers showing giant germ-line nurse cells surrounded by E-cadherin-positive follicle cell epithelium. A follicle cell undergoing mitotic rounding is highlighted with an asterisk (*). Scale bar ~3 μm. *n* > 3 independent biological replicates.

B   High-resolution imaging of the F-actin cortex, E-cadherin and phospho-MyoII in a mitotically rounded follicle cell. Note maintenance of adherens junctions including both E-cad and the contractile actomyosin ring during mitotic rounding. Asterisks or pH3 staining indicates a single mitotic cell. Scale bar ~3 μm. *n* > 4 independent biological replicates.

C   Schematic diagram of follicle cell rounding, showing the relatively mild shape change that is insufficient to affect E-cadherin levels during mitotic rounding.

E-cadherin in the pseudostratified wing imaginal disc epithelium of *Drosophila*. Our findings are consistent with previous observations that adherens junctions can be removed via E-cadherin endocytosis upon planar polarised Rho activation and actomyosin-driven junctional shrinkage generated during cell–cell rearrangements in the *Drosophila* embryo [47,48] and recent optogenetic experiments in human cells [82]. However, the global loss of E-cadherin we observe during mitosis of pseudostratified epithelial cells is unprecedented and may be uniquely required by the rapid transformation of these cells from their highly columnar shape to a rounded sphere at mitosis, which involves a rapid increase in apical area and junctional length. This change in cell chape, driven by global actomyosin contractility during mitotic rounding, may both spread out junctions and disrupt cadherin–cadherin contacts between neighbouring cells to favour endocytosis.

Notably, loss of E-cadherin does not seem to occur during mitosis in cuboidal epithelial cells, which undergo a much milder cell shape change during mitotic rounding, such as the *Drosophila* follicle cell epithelium or in many human cultured epithelial cell lines. Indeed, E-cadherin was reported to play essential roles in planar spindle orientation in cultured human cells [44–46]. Thus, the pseudostratified epithelia of *Drosophila* face a unique challenge of orienting the mitotic spindle in the plane of the epithelium without the use of adherens junctions as a cue, which may explain why these cells instead rely on septate junctions [21], while planar spindle orientation can occur normally in the cuboidal follicle cell epithelium before septate junctions form. In the absence of septate junctions, spindle-orienting factors such as Pins, Mud, Dlg and Scrib localise to lateral membranes, overlapping with E-cadherin [20], which can directly interact with Scrib in both *Drosophila* and human cells [83,84].

Our findings also shed light on the molecular mechanisms linking the cell cycle with control of Rho activation during mitotic rounding. Downstream of the master mitotic kinase Cdk1, we identify a key role for both Aurora A and B kinases (acting redundantly) in initiating mitosis and maintaining cortical rounding, with Aurora B then acting alone to drive furrow formation during cytokinesis. Aurora kinases are known to activate the key cell cycle kinase Plk1/Polo [64,65], which can then activate RacGAP1/MgcRacGAP/CYK4/Tum and ECT2/Pbl [61,62], possibly via the kinesin-like protein MKLP1/KIF23/ZEN4/Pav [85]. One report suggested that Aurora B also acts to directly phosphorylate RacGAP1/MgcRacGAP/CYK4/Tum on S387 [86], but we find that a CRISPR-knockin mutation of this site ($tum^{S387A}$) is homozygous viable and fertile in *Drosophila*. In addition, Aurora B can also directly phosphorylate MKLP1/KIF23/ZEN4/Pav to oligomerise and activate RacGAP1/MgcRacGAP/CYK4/Tum and ECT2/Pbl during cytokinesis [73–76], via a mechanism involving plasma membrane association of clustered C1 domains from the RacGAP1/MgcRacGAP/CYK4/Tum protein [77,78]. Accordingly, we find that a CRISPR-knockin mutation of this site ($pav^{S734A/S735A}$) is homozygous lethal in *Drosophila*, suggesting that Aurora A/B could act via this complex during mitotic rounding. Indeed, MKLP1/KIF23/ZEN4/Pav undergoes cell cycle-dependent re-localisation from the nucleus (in interphase) to the cytoplasm (in mitosis), including a clear localisation to the entire plasma membrane during mitosis and then to the cleavage furrow during cytokinesis [87]. Our results show that mitotic activation of the MKLP1/KIF23/ZEN4/Pav binding partner RacGAP1/MgcRacGAP/CYK4/Tum also involves translocation from the nucleus to the cytoplasm, similar to ECT2/Pbl [34], where all three proteins are then available to bind the entire plasma membrane and generate global Rho activity to drive cortical contractility and loss of adherens junctions during mitotic rounding.

The complex of MKLP1/KIF23/ZEN4/Pav and RacGAP1/MgcRacGAP/CYK4/Tum is often referred to as the "centralspindlin" complex, due to its association with the spindle midzone at anaphase [54]. However, we have avoided the use of this term because association with the spindle midzone is not required for the function of this complex in activating ECT2/Pbl and Rho at the plasma membrane during cytokinesis [88]. Furthermore, our results show that the same complex also functions prior to anaphase or cytokinesis in activating ECT2/Pbl and Rho for cortical rounding and downregulation of adherens junctions from the very onset of mitosis.

Importantly, the function of the three cell cycle-regulated Rho activators discussed here—MKLP1/KIF23/ZEN4/Pav, RacGAP1/MgcRacGAP/CYK4/Tum and ECT2/Pbl—on the regulation of adherens junctions appears to be dose-dependent and cell shape change-dependent. The sudden re-localisation of these proteins to the cytoplasm at mitosis drives a global increase in Rho activation, cell shape change and loss of junctions. In contrast, the relatively low level of these proteins in the cytoplasm during interphase appears to contribute to the normal maintenance of Rho activity at the junctional actomyosin ring to maintain adherens junctions—at least in mammalian cells [89]. In *Drosophila*, an interphase role of ECT2/Pbl in contributing to Rho activation and maintenance of adherens junctions is also plausible and should be most easily distinguished from the mitotic role in non-dividing epithelial cells. However, in the ovarian follicle cell epithelium, silencing of ECT2/Pbl by RNAi affected cytokinesis (leading to larger cells) but did not affect overall epithelial architecture after cells arrest their proliferation, suggesting that any interphase function may be obscured by redundancy with other RhoGEFs in this tissue (Fig EV3). Note also that, in this cuboidal epithelium, mitotic rounding itself is more subtle and ECT2/Pbl is dispensable for planar spindle orientation and epithelial integrity (Figs 8 and EV3).

Finally, we note that ECT2/Pbl was initially reported to be capable of acting as GEF for another GTPase, Cdc42, in addition to Rho [38]. This activity suggested a possible role in regulating the apical Cdc42-Par6-aPKC complex in epithelial polarity [90] and in mitosis [91]. In *Drosophila*, ECT2/Pbl was found to drive transient apical spreading of the Cdc42-Par6-aPKC complex during mitotic rounding of pupal notum epithelial cells [36]. We find evidence for a similar function of ECT2/Pbl in activating Cdc42 during interphase in the post-mitotic ovarian follicle cell epithelium, although the loss-of-function phenotype of ECT2/Pbl is obscured due to redundancy with other Cdc42 GEFs such as beta-PIX (Fig EV3). Loss of both ECT2/Pbl and beta-PIX reduced the level of the Cdc42-Par6-aPKC complex localising to the apical domain of post-mitotic follicle cells (Fig EV3) and also reduced apical ZO-1 localisation in human Caco2 intestinal epithelial cells in culture (Fig EV4). In contrast, the gain-of-function phenotype caused by ECT2/Pbl overexpression is a neoplastic tumour-like phenotype and clearly involves ectopic spreading of the apical Cdc42-Par6-aPKC complex, in addition to persistent cell rounding in both

cuboidal follicle cells and pseudostratified wing epithelial cells (Figs EV5 and EV6).

In conclusion, our findings provide new insights into the cell biology of mitotic rounding, identifying remodelling of adherens junctions as a key event in pseudostratified epithelia, where rounding is extensive, but not cuboidal epithelia, where rounding is more subtle. These results are consistent with the hypothesis that Rho activation and actomyosin contractility can stabilise adherens junctions in the absence of mechanical strain, but that Rho activation can induce E-cadherin endocytosis above a critical strain threshold [47,82], be it either junctional shrinkage or expansion, both of which may alter the geometry of the junctional actomyosin ring and disrupt cadherin–cadherin contacts between neighbouring cells to favour endocytic internalisation of E-cadherin. Our results also clarify the molecular mechanisms linking cell cycle control machinery, particularly the Aurora A and B kinases, with Rho activation and mitotic rounding. Lastly, our work may have direct relevance to certain human epithelial cancers, such as lung cancer or glioma, where overexpression of ECT2/Pbl has been reported to correlate for poor prognosis [92,93], and where our findings suggest it could drive not only disruption of epithelial polarity via activation of Cdc42 or Rac [94,95] but also loss of adherens junctions via sustained Rho activation to promote tumour progression.

# Materials and Methods

### *Drosophila* genetics

All stocks are described in FlyBase and were obtained from the Bloomington *Drosophila* Stock Centre or the Vienna *Drosophila* RNAi Centre. Transgenes driven by a UAS promoter were induced by crossing lines to the whole wing-specific *MS1096.Gal4* driver line or P-compartment-specific *hh.Gal4* driver line. Clones expressing *UAS.GFP* in either homozygous *FRT* wild-type or *FRT* mutant backgrounds were generated using the MARCM method for *FRT*-site-directed mitotic recombination by heatshocking flies at 37 degrees for 1 h during early larval development to induce the FLP recombinase expression from a *hs.flp* transgene and waiting until third larval instar before dissection.

### Antibody staining

Wing imaginal discs were dissected from third instar larvae and fixed for 30 min in cold PBS containing 4% paraformaldehyde. Discs were washed in PBS supplemented with 0.1% Triton X-100 (PBT), blocked with 0.1% bovine serum albumin (PBT + BSA), and stained with primary and fluorescently conjugated secondary antibodies.

Primary antibodies used were as follows: Rabbit anti-Phospho Myosin-II Regulatory Light Chain (1:50, Cell Signaling), rabbit anti-Phospho LLGL1/2 (1:250, Abgent), rat anti-DE-Cadherin (1:100, DSHB), mouse anti-pH3 (1:1,000, Abcam) and rabbit anti-phospho-histone H3 (1:1,000, Millipore).

Secondary antibodies (all from Molecular Probes, Invitrogen) were used at 1:500 for 2 h prior to multiple washes in PBT and staining with DAPI at 1 μg/ml for 10 min, before mounting on slides

in Vectashield (Vector labs). Images were taken on a Leica SP5 confocal and processed with Adobe Photoshop.

### Live imaging

Live imaging of *ex vivo* cultured wing discs was performed with a Zeiss 780 confocal microscope. Briefly, discs were cultured in Shields and Sang M3 media (Sigma) containing 2% foetal bovine serum, 10 μg/ml streptomycin/penicillin (Invitrogen), 10 mU/l insulin, 0.1 μg/ml ecdysone (Sigma) and 2.5% methyl cellulose (Sigma), and imaged in a 35-mm FluoroDish. *Z*-stacks were taken at 1-μm intervals, with total thickness of 10–30 μm. *Z*-stacks were typically scanned at 2-min intervals for up to 3 h. Images were projected, and time points were collated using Zen software.

For drug treatments, inverted larvae were incubated in culture media as described above (without methyl cellulose) with the respective drug for 30–60 min and then transferred to media as described above for imaging. Drugs used were 2 mM VX-680/Tozasertib (Selleck BioChem), 0.1 mg/ml colchicine (Sigma, C9754) and 2.5 mM Y-27632 (Millipore).

### Transmission electron microscopy

Imaginal discs were dissected in 4% PFA in PBS and transferred onto coverslips coated with poly-L-lysine. Discs were further fixed for 2 h in 4% PFA + 2.5% glutaraldehyde in 0.1 M phosphate buffer pH 7.4 (PB) at room temperature. Discs were then postfixed in 1% osmium/1.5% potassium ferrocyanide for 1 h, followed by 1% tannic acid in 0.05 M sodium cacodylate pH 7.4 for 45 min. Coverslips were dehydrated stepwise through ethanol, infiltrated with 50:50 propylene oxide: epon followed by one change of pure resin every 24 h for 7 days and then polymerised at 60°C overnight. Ultrathin sections of ~75 nm were collected using a UCT ultramicrotome (Leica Microsystems UK), post-stained with lead citrate and viewed using a Tecnai G2 Spirit 120 kV transmission electron microscope (FEI Company) with an SC1000 Orius CCD camera (Gatan UK).

**Expanded View** for this article is available online.

### Acknowledgements

We thank Ken Blight in the Electron Microscopy facility for help with preparation of EM images. We thank Maria del Carmen Diaz de la Loza for assistance with one of the follicle cell immunostainings as well as for improving the diagrams. We also thank Ahmed Elbediwy for assistance with the Caco2 cell experiments. This work was funded by the Francis Crick Institute (FC0010180), which is a joint initiative of Cancer Research UK (FC0010180), the Medical Research Council (FC001180), and Wellcome (FC001180). This work was also funded by The Australian National University.

### Author contributions

MA-A, TTB, GPB and GCF performed the experiments and analysed the data. BJT conceived the experiments and wrote the manuscript with input from the other authors.

### Conflict of interest

The authors declare that they have no conflict of interest.

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
