## [Review Process File · EMBO Reports]

Adherens junction remodelling during mitotic rounding of pseudostratified epithelial cells

Mario Aguilar-Aragon, Teresa T Bonello, Graham P Bell, Georgina C Fletcher & Barry J Thompson

Review timeline:

Submission date:	18 November 2019
Editorial Decision:	27 November 2019
Revision received:	10 December 2019
Accepted:	15 January 2020

Editor: Deniz Senyilmaz-Tiebe

Transaction Report: This manuscript was transferred to *EMBO reports* following peer review at *The EMBO Journal*

Transfer

18 November 2019

Referee #1:

In this study, the authors address the important question how cells undergo mitotic rounding in the crowdedness of a growing epithelium, focusing on a specialized epithelial type, the pseudostratified epithelium. They claim that cells in these epithelia downregulate cadherin while rounding and that this depends on Rho kinase mediated actomyosin contractility, differently from what occurs in cuboidal epithelia. As pseudostratified epithelia are common in many developing tissues but their cell biology the molecular mechanisms underlying cell biological features are so far understudied, this work is a potentially important contribution. While many initial observations in this study are interesting, some of them fall short in the depth of experimental design and insight. Further, some parts of the manuscript seem sloppy and are in need for improvement (scale bars!!). On the other hand, the authors tend to overstate some of their findings. Thus, more controls and other experiments should be performed before this manuscript is ready for publication. Alternatively, the authors would need to tune down some of their claims.

Major points

- The authors conclude their abstract by saying that 'mitosis in pseudostratified epithelia necessitates planar spindle orientation by septate junctions to maintain epithelial integrity' However, this statement is not yet verified in their data. More experiments need to be done to justify this statement or it needs to be removed (see below).

> **We thank the reviewer for raising this important point. We have now modified the language used, to specify that we are talking about *Drosophila*: “Thus, in *Drosophila* pseudostratified epithelia, disruption of adherens junction during necessitates planar spindle orientation by septate junctions to maintain epithelial integrity”. Further work is need to investigate whether E-cadherin is also reduced during mitosis in mammalian pseudostratified epithelia, and if so, whether these cells would then require other junctions such as tight junctions to orient the spindle.**

- Figure 2, do cells delaminate upon excessive Rock inhibitor treatment? If not, how could this be explained?

> Since the entire disc is treated for several hours, the tissue slowly loses integrity of junctions and ends up quite abnormal (with delamination of cells), but is difficult to analyse in these movies. We therefore do not draw any specific conclusions except that E-cadherin is strongly reduced under these conditions.

- Can the authors comment on what happens to mitosis itself in case of Rho dependent contractility inhibition and in case cadherin stays at the junctions?

> This is an interesting point. In our experiments, treatment with Rock-inhibitor appears to trap cells in a rounded state, as if the cell is still 'trying to round up and complete mitosis and cytokinesis'. Existing literature suggests that this is because complete rounding is required for formation of a proper spindle, such that Rock-inhibitor treatment can lead to a spindle assembly checkpoint delay (see for example Buzz Baum's and Manuel Thery's work cited in our manuscript).

- The interpretation of Figure 4 does not match the presented data. No extrusion is shown for either mutant tissue nor are problems with spindle orientation or mitotic rounding clear from the Panels. This data needs to be added or the statements need to be tuned down.

> We agree. To address this point, we modify the text to emphasise that we are using Dcp1-positive apoptotic cells as a proxy for extruded cells. Similarly, we now emphasise that we are using p-MyoII as a proxy for proper mitotic rounding.

- Figure 7C, how do the authors explain the upregulation of Myosin 2 without downregulation of Cadherin in cells that had already entered mitosis when the drug was applied, as stated in text?

> To clarify, these cells were arrested mid-way through rounding and so are not 'rounded enough' (don't undergo sufficient cell shape change) to downregulate E-cadherin. This supports our hypothesis that mechanical strain (shape change), rather than stress (MyoII-mediated tension), is the key factor in downregulating E-cadherin.

- The authors conclude a lot about the role of 'mechanical strain' in their phenomenon. However, no experiment has really addressed the mechanics of the system, as so far all experiments are molecular inhibitions and depletions. In this context more experiments need to be added or it has to be clarified that Figure 6D is a hypothesis that has not yet been confirmed (on the mechanical strain part). The same applies when the authors refer to 'mechanosensitive junction remodeling' in the discussion.

> We accept this comment. We have now clarified the text to explain that this figure (revised 7D) represents our hypothesis that cell shape change (mechanical strain) is causing downregulation of E-cadherin. We also soften our language and specifically reduce the use of the terms 'mechanical strain' and 'mechanosensitive' in the revised manuscript.

Figure comments

- All Figures lack scale bars! That is hardly acceptable for a manuscript sent to a journal for review. Scale bars need to be added to all Figures and need to be explained in legends.

> Fixed.

- The schemes seem a bit underdeveloped and could be aesthetically improved.

> We have now updated the graphical abstract.

- All bar graphs should be replaced by graphs showing data points as for example done in Figure 3A and all of Figure 4 according to current state of the art.

- Figure 2B and C lower panels are very pixilated and should be replaced. Same for Figure 3B.

> We apologise for this, which is caused by our imaging of whole discs, and then zooming in on single cells. Unfortunately, since my lab is currently moving to Australia, I don't have access to the slides to re-image. Instead, we hope the quantification provided is sufficient.

- Figure 3B, graph, what does 100% refer to?

> To clarify, the p-MLC intensity was measured in ImageJ and then expressed as a percentage of the maximum measured intensity observed.

Minor comments

- The material and methods section is very sparse. More details on experiments and statistics could be added.

Fixed.

- In the manuscript the authors jump a lot between Figure 4 and 3 which makes it hard to follow the flow and Figures, this should be reshuffled.

Fixed.

- Figure 3C is never mentioned in the text...???

Fixed.

Referee #2:

In this manuscript Aguilar-Aragon et al delineate a molecular pathway controlling cell shape changes during mitosis (rounding) in pseudostratified *Drosophila* epithelial cells. They demonstrate that activation of actomyosin contractility leads to E-cadherin downregulation and adherens junction disassembly, presumably through mechanical forces. They further dissect the molecular pathway linking the cell cycle machinery with mitotic rounding which they show requires Aurora A/B kinase activation of the RhoGEF (Pbl). They speculate that this might explain why spindle orientation in pseudostratified epithelia requires septate junction but not adherens junctions. Consistently, they show that in simple epithelial cells, such as cuboidal follicle cells in the *Drosophila* ovary, which undergo only moderate cell shape changes during mitosis, adherens junctions do not disassemble. Collectively these results provide molecular explanation of how epithelial cells with complex shape divide without compromising tissue integrity.

The experiments are well-controlled and the results, which are based on a combination of *Drosophila* genetics, pharmacological perturbations and imaging, support the main conclusions of the manuscript. The results could be further strengthened if the authors could show that acute activation of actomyosin contractility leads to cadherin disassembly by using for example optogenetics. However, I do appreciate that this experiment might be tricky to implement in the wing disk.

> This is a really interesting suggestion, although we are unfortunately not set up to perform this experiment in wing discs for technical reasons, although this might form the basis for a long-term future project on optogenetics.

While I very much like the experimental design undertaken, I find that the text requires some modifications. Several terminologies appear to be not fully justified given the data shown. In particular the use of "mechanosensitive" in the title, abstract and throughout the manuscript should be replaced with something like "mechanical forces". Mechanosensitive implies sensing mechanisms (see for example talin, integrin or piezo) which have not been here demonstrated.

> This is an important point. In the revised manuscript, we have avoided the use of the term 'mechanosensitive' in the title, abstract and elsewhere, as suggested.

Also, there are several occasions where the authors refer to mechanical strain and stress which have not been measured. The parallel between cell shape changes and mechanical strain in this context seems preliminary as cell shape changes can occur also by non-mechanical mechanisms (modulation of endo/exocytosis or membrane remodeling). I suggest to stick to a more generic "cell shape changes" terminology and clearly indicate when specific terminology is used in a speculative manner. For example, see p. 9 " This dramatic increase in mechanical strain (shape change) driven by increased mechanical stress (global contractility) during mitotic rounding may disrupt cadherin-cadherin contacts between neighbouring cells to favour endocytosis".

> We accept this point. We have now primarily use 'cell shape changes' rather than 'mechanical strain' in the revised manuscript.

Lastly please avoid the excessive use of non-quantitative adjectives such as "dramatic/strong/very" as they disturb the reading of the manuscript see for example p.6 "The plasma membrane localisation of GFP-Tum during mitosis is not as strong as that found at the cytokinetic furrow or at ring canals formed after cytokinesis, but is nevertheless much stronger than that occurring during interphase".

> A very good point. We have now altered the language accordingly.

1st Editorial Decision

27 November 2019

Thank you for transferring your manuscript to EMBO Reports. Your manuscript has been reviewed and revised at another journal. Having looked at everything carefully, I would like to invite a minor revision, before I can accept the manuscript.

1st Revision - authors' response

10 December 2019

The authors performed all minor editorial changes.

Corresponding Author Name: Barry Thompson

Journal Submitted to: EMBO reports

Manuscript Number: EMBOR-2019-49700